# Cerebellar Cortex and Cerebellar White Matter Volume in Normal Cognition, Mild Cognitive Impairment, and Dementia

**DOI:** 10.3390/brainsci11091134

**Published:** 2021-08-26

**Authors:** Nauris Zdanovskis, Ardis Platkājis, Andrejs Kostiks, Oļesja Grigorjeva, Guntis Karelis

**Affiliations:** 1Department of Radiology, Riga Stradins University, Dzirciema Street 16, LV-1007 Riga, Latvia; ardis.platkajis@rsu.lv; 2Department of Radiology, Riga East University Hospital, Hipokrata Street 2, LV-1038 Riga, Latvia; 3Department of Neurosurgery and Neurology, Riga East University Hospital, Hipokrata Street 2, LV-1038 Riga, Latvia; andrejs.kostiks@gmail.com (A.K.); guntis.karelis@gmail.com (G.K.); 4Department of Computer Control Systems, Riga Technical University, Kaļķu Street 1, LV-1658 Riga, Latvia; Olesja.Grigorjeva@rtu.lv

**Keywords:** cerebellum, cortex, volumetry, MRI, dementia, cognitive impairment, white matter

## Abstract

The cerebellum is commonly viewed as a structure that is primarily responsible for the coordination of voluntary movement, gait, posture, and speech. Recent research has shown evidence that the cerebellum is also responsible for cognition. We analyzed 28 participants divided into three groups (9 with normal cognition, 9 with mild cognitive impairment, and 10 with moderate/severe cognitive impairment) based on the Montreal Cognitive Assessment. We analyzed the cerebellar cortex and white matter volume and assessed differences between groups. Participants with normal cognition had higher average values in total cerebellar volume, cerebellar white matter volume, and cerebellar cortex volume in both hemispheres, but by performing the Kruskal–Wallis test, we did not find these values to be statistically significant.

## 1. Introduction

The cerebellum is commonly seen as a structure that is primarily responsible for the coordination of voluntary movement, gait, posture, and speech [1,2,3]. Recent research has shown there is clear evidence that the cerebellum is responsible not only for motor functions but also for cognition [1,4].

Based on cerebellar anatomy, it is not difficult to theoretically conclude that the cerebellum could be involved in cognition; i.e., we have approximately 16 billion neurons in the cerebral cortex compared to 69 billion neurons in the cerebellum, and the same analogy applies to cerebellar connections [1,5]. The cerebellum is connected to the brain via three peduncles (superior, middle, and inferior cerebellar peduncles), which form cerebellar pathways and connections to the brain. There are several afferent tracts (*cortico-ponto-cerebellar*, *cortico-olivo-cerebellar*, *cortico-reticulo-cerebellar*, *spinocerebellar*, *dorsal spinocerebellar*, *ventral spinocerebellar*, *cuneocerebellar*, *vestibulocerebellar*, *rostral spinocerebellar tract*, *and vestibulocerebellar tract*) and efferent tracts (*rubrospinal tract and cerebellovestibular tract*) [5,6]. These tracts connect to the sensorimotor and association areas of the cerebral cortex [7]. In functional magnetic resonance studies (fMRI), the functional topography of the cerebellum shows that sensorimotor tasks engage the cerebellar anterior lobe and lobule VIII, and cognitive tasks activate the posterolateral cerebellar hemispheres [8,9,10]. Neuroimaging studies suggest that the cerebellum forms anatomical connections with the prefrontal cortex, which is important for normal cognitive function [11,12,13]. Additionally, there is evidence that abnormal prefrontal–cerebellar connections are seen in patients with autism [14,15] and schizophrenia [16].

In terms of cerebellar contribution to cognition, several mechanisms have been described. In particular, ***cerebellar cognitive affective syndrome (CGAS)*** is characterized by impaired executive function (including planning, set-shifting, abstract reasoning, verbal fluency, working memory), visuospatial memory and perception impairment, personality changes, and impaired verbal fluency. These impairments were seen in patients with large, bilateral, or pan-cerebellar disorders. Cerebellar lesions were located in the posterior lobe, vermis, and anterior lobe [1,17,18].

In ***cerebellar stroke,*** lesion location is associated either with motor or cognitive deficits; i.e., anterior lobe lesions usually impact motor functions, and posterior lobe lesions are associated with worse cognitive scores [8,10,18].

It is important to note ***cerebellar changes with aging*; i.e.,** with aging, there is a 10% to 40% decrease in the Purkinje cell layer and reduction in dorsal vermis volume [1,19,20]. While the cerebellum is not the primary region of interest in aging and cognition, cell loss in this region could potentially lead to functional changes [1].

In recent studies, there is evidence that the cerebellum contributes to cognitive functioning, and, thus, it should be noted when analyzing magnetic resonance examinations in patients with cognitive impairment.

In our study, we evaluated cerebellar white matter volume and cerebellar cortex volume in participants with normal cognition, mild cognitive impairment, and moderate/severe cognitive impairment.

## 2. Materials and Methods

### 2.1. Participant Groups and Montreal Cognitive Assessment Cutoff Scores

Participants with suspected cognitive impairment were admitted to the neurologist. All participants were evaluated by the board-certified neurologist, and the Montreal Cognitive Assessment (MoCA) was carried out. All participants had at least 16 years of higher education. We divided participants into 3 groups [21,22]:1.***Participants with normal cognition (NC)*** with MoCA scores ≥27;2.***Participants with mild cognitive impairment (MCI)*** with MoCA scores ≥18 and ≤23;3.***Participants with moderate and severe cognitive impairment (SCI)*** with MoCA scores ≤17.

We excluded participants with MoCA scores of 24, 25, and 26 as these scores are inconclusive and can provide false-positive or false-negative results. Scores lower than 23 are not healthy with 88% accuracy and scores ≥27 are not pathological with 91% accuracy [23]. Participant demographic data and MoCA results based on the participant group are shown in Table 1.

Exclusion criteria for participants were clinically significant neurological diseases (tumors, major stroke, malformations, etc.), drug use, and alcohol abuse. Study participants did not have other significant pathological findings on magnetic resonance imaging (MRI).

Based on the neurological assessment and MRI data, participants in MCI and SCI groups had mixed types of cognitive impairment; i.e., we did not have participants in MCI and SCI groups that had clear evidence of one type of dementia (vascular dementia, Alzheimer’s disease, Lewy body dementia, or frontotemporal dementia). All patients had at least some T2 white matter hyperintensities and some degree of global cortical atrophy in combination with other cerebral lobe atrophies.

### 2.2. Magnetic Resonance Imaging (MRI) Data Acquisition and Post-Processing with Freesurfer

MRI scans were performed on a single site 3T scanner in a university hospital setting. For cerebellar parcellation, we used a sagittal 3D T1-weighted MP RAGE (Magnetization Prepared Gradient Echo) sequence with 1 mm voxel size. Scans were converted from DICOM format to neuroimaging informatics technology initiative (NIfTI) files and then post-processed. Cortical reconstruction and volumetric segmentation were performed with the FreeSurfer 7.2.0. image analysis suite, which is documented and freely available for download online (http://surfer.nmr.mgh.harvard.edu/, accessed on 1 July 2021). The technical details of FreeSurfer processing steps and procedures are described in prior publications [24,25,26,27,28,29,30,31,32,33,34,35,36,37,38,39,40,41,42].

### 2.3. Statistical Analysis

Statistical analysis was performed by using the software JASP Version 0.14.1 (Amsterdam, Netherlands). We analyzed descriptive statistics in all participant groups and performed the Kruskal–Wallis test to identify statistically significant differences between groups. Furthermore, we analyzed descriptive statistics for each participant group for both cerebellar white matter volume and cortex volume, including mean, median, standard deviation, minimum, and maximum values.

## 3. Results

We compared total cerebellum volume, white matter volume and cortex volume in both cerebellar hemispheres and performed the Kruskal–Wallis test to evaluate statistical significance between groups. In addition, we calculated mean, median, standard deviation, minimum, and maximum values for each participant group. Results are grouped based on cerebellar anatomical parcellation—left hemisphere white matter and cortex volume, right hemisphere white matter and cortex volume, and total cerebellum volume.

### 3.1. Left Cerebellum White Matter Volume

Left cerebellum white matter volume data were acquired from FreeSurfer parcellation statistical data. The descriptive statistics of participant groups are shown in Table 2.

Comparing mean values in groups, we found the highest values in the normal cognition participant group and the lowest values were found in the mild cognitive impairment group (Figure 1).

The Kruskal–Wallis test was conducted to assess differences between groups. No statistically significant changes were found between participant groups (H (2) = 3.476, *p* = 0.176).

### 3.2. Left Cerebellum Cortex Volume

Left cerebellum cortex volume data were acquired from FreeSurfer parcellation statistical data. The descriptive statistics of participant groups are shown in Table 3.

Comparing mean values in the groups, we found the highest values in the normal cognition participant group, the lowest values were found in the mild cognitive impairment group, and the highest variability was found in the moderate/severe cognitive impairment group (Figure 2).

The Kruskal–Wallis test was conducted to assess differences between groups. No statistically significant changes were found between participant groups (H (2) = 0.970, *p* = 0.616).

### 3.3. Right Cerebellum White Matter Volume

Right cerebellum white matter volume data were acquired from parcellation statistical data. The descriptive statistics of participant groups are shown in Table 4.

Comparing mean values in groups, we found the highest values in the normal cognition participant group, and the lowest values were found in the moderate/severe cognitive impairment group (Figure 3).

The Kruskal–Wallis test was conducted to assess differences between groups. No statistically significant changes were found between participant groups (H (2) = 2.987, *p* = 0.225).

### 3.4. Right Cerebellum Cortex Volume

Right cerebellum cortex volume data were acquired from FreeSurfer parcellation statistical data. The descriptive statistics of participant groups are shown in Table 5.

Comparing mean values in groups, we found the highest values in the normal cognition participant group, and the lowest values were found in the mild cognitive impairment group (Figure 4).

The Kruskal–Wallis test was conducted to assess differences between groups. No statistically significant changes were found between participant groups (H (2) = 0.467, *p* = 0.792).

### 3.5. Total Cerebellum Volume

Total cerebellum volume was calculated as a sum of white matter and gray matter volume in the right and left cerebellum. The descriptive statistics of participant groups are shown in Table 6.

Comparing mean values in groups, we found the highest values in the normal cognition participant group, and the lowest values were found in the mild cognitive impairment group (Figure 5).

The Kruskal–Wallis test was conducted to assess differences between groups. No statistically significant changes were found between participant groups (H (2) = 1.188, *p* = 0.552).

## 4. Discussion

In our study, we focused on general cerebellar quantitative measurement, i.e., white matter and cortical volume in both cerebellar hemispheres.

Today, the cerebellum is recognized as an associative center of higher cognitive functions [43,44]. Thanks to the fMRI studies, it was possible to map functional cerebro-cerebellar connections and identify specific cerebellar regions that are responsible for cognition [45].

Although the cerebellum contributes to cognition, when correlating cerebellar size, volume, or general structure, it has a poor correlation with cognitive tests. These findings have been reported in several articles:

Paradiso et al. correlated cerebellar volume with general intelligence, and although there was a positive correlation, it was not statistically significant [46].

Bernard et al. analyzed the volume of several anatomical regions in the cerebellum (anterior part, crus 1, posterior part, and vermis) and correlated volumetric data with cognitive tasks (digit symbol, trails, spatial span, spatial learning, letter span, and verbal learning) and did not find a statistically significant relationship [47].

Hoogenda et al. found a minor relationship between larger cerebellar volume and better cognition in healthy older adults, which further attenuated after correcting for cerebral volume, concluding that it is not the main leading structure in terms of cognition [48].

Mitoma et al. in a consensus paper discussed cerebellar reserve and cerebellar cognitive reserve. The cerebellar reserve is defined as the capacity of the cerebellum to compensate and restore function in response to pathology. Regarding cerebellar cognitive reserve, there may be no structural or functional differences in different disease groups (i.e., Alzheimer’s disease, frontotemporal dementia, autism spectrum disorder, schizophrenia, and major depressive disorder) rather than compensatory reorganization changes that improve behavior and cognition [49,50].

In general, although we did not achieve statistically significant results in our study, it is important to be aware of cerebellar contribution to cognitive function, especially in diagnostic imaging in patients with cognitive impairment.

Further research is necessary to assess more detailed cerebellar anatomical volumetric measurements and correlate those findings with clinical data and cognitive testing.

### Limitations

This was an exploratory study to assess the cerebellar white matter and cortical volume in association with cognitive impairment and dementia. The limitations of this study include cross-sectional design and a small participant cohort. In addition, it is necessary to note that there were significant differences in the mean age of the participants in study groups. The mean age for the NC group was younger than that for the MCI and SCI groups. Thus, we expected the differences between the NC group and either the MCI group or SCI group to be statistically significant, but we did not find any statistically significant differences.

## 5. Conclusions

In our study, we did not find statistically significant differences in cerebellar cortex volume and cerebellar white matter volume in participant groups with normal cognition, mild cognitive impairment, and dementia.

## Figures and Tables

**Figure 1 brainsci-11-01134-f001:**
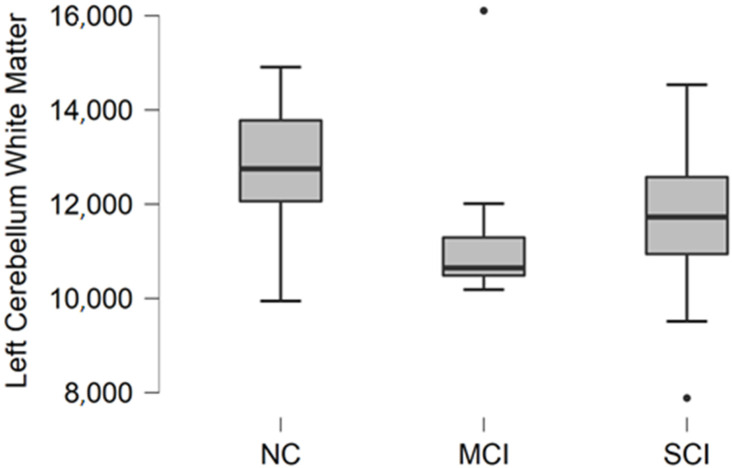
Mean left cerebellum white matter volume in mm^3^ with standard deviation in participant groups. Black dots represent outliers (i.e., exceptionally diverging values).

**Figure 2 brainsci-11-01134-f002:**
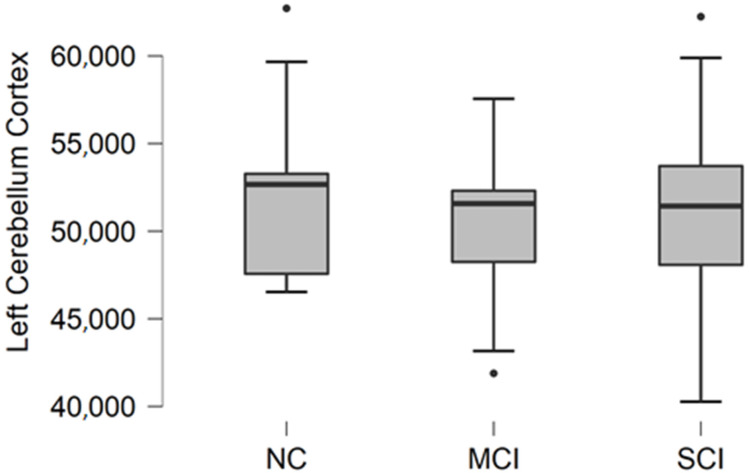
Mean left cerebellum cortex volume in mm^3^ with standard deviation in participant groups. Black dots represent outliers (i.e., exceptionally diverging values).

**Figure 3 brainsci-11-01134-f003:**
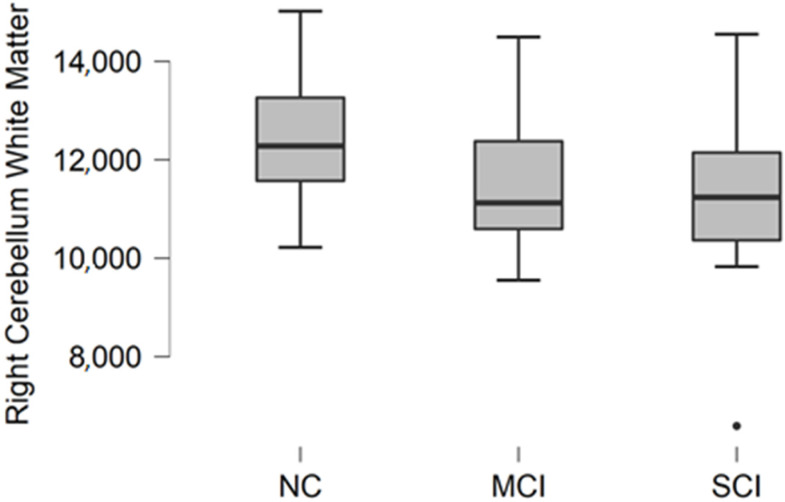
Mean right cerebellum white matter volume in mm^3^ with standard deviation in participant groups. Black dots represent outliers (i.e., exceptionally diverging values).

**Figure 4 brainsci-11-01134-f004:**
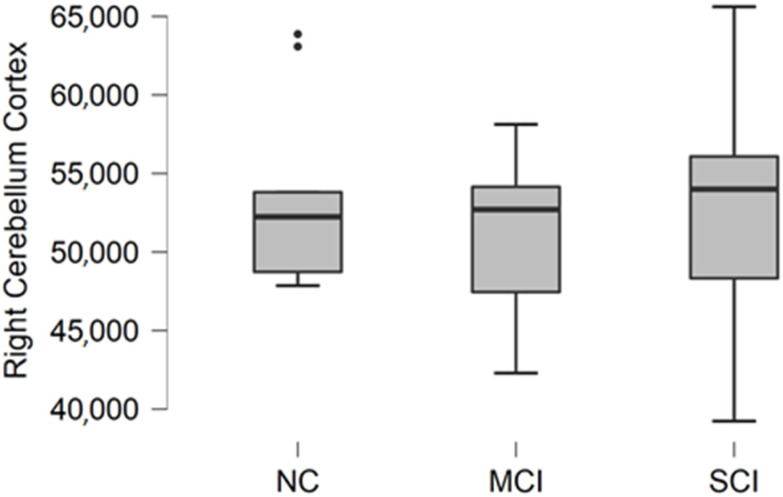
Mean right cerebellum cortex volume in mm^3^ with standard deviation in participant groups. Black dots represent outliers (i.e., exceptionally diverging values).

**Figure 5 brainsci-11-01134-f005:**
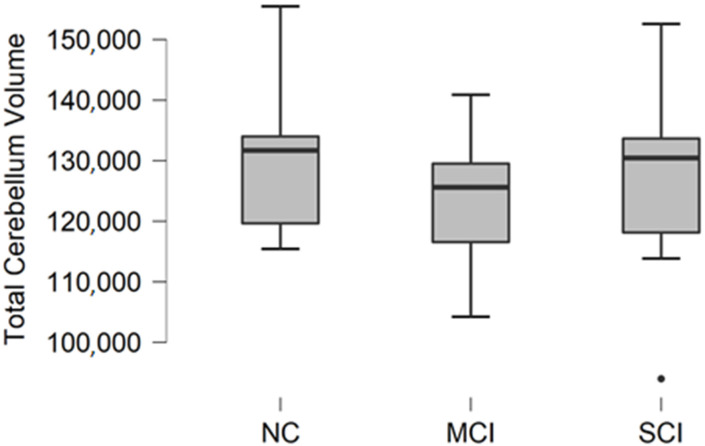
Mean total cerebellum volume values in mm^3^ with standard deviation in participant groups. Black dots represent outliers (i.e., exceptionally diverging values).

**Table 1 brainsci-11-01134-t001:** Demographic data and MoCA scores in participant groups (NC—normal cognition, MCI—mild cognitive impairment, SCI—moderate and severe cognitive impairment).

	Age			MoCA		
	NC	MCI	SCI	NC	MCI	SCI
Participants	9	9	10	9	9	10
Female:Male	6:3	4:5	6:4	6:3	4:5	6:4
Mean	55.1	71.4	74.8	28.7	20.9	10.1
Std. Error of Mean	5.4	2.4	3.3	0.3	0.7	1.3
Median	51.0	71.0	71.0	29.0	20.0	11.5
Std. Deviation	16.1	7.2	10.4	1.0	2.1	4.1
Minimum	35.0	58.0	66.0	27.0	18.0	4.0
Maximum	76.0	82.0	96.0	30.0	23.0	15.0

**Table 2 brainsci-11-01134-t002:** Descriptive statistics for left cerebellum white matter volume in participants with normal cognition (NC), mild cognitive impairment (MCI), and moderate/severe cognitive impairment (SCI).

	Left Cerebellum White Matter Volume, mm^3^		
	NC	MCI	SCI
Participants	9	9	10
Mean	12,684.2	11,429.8	11,506.7
Median	12,750.5	10,647.7	11,726.8
Std. Deviation	1699.8	1849.2	1852.4
Minimum	9944.9	10,186.9	7885.2
Maximum	14,908.2	16,105.9	14,536.1

**Table 3 brainsci-11-01134-t003:** Descriptive statistics for left cerebellum cortex volume in participant groups with normal cognition (NC), mild cognitive impairment (MCI), and moderate/severe cognitive impairment (SCI).

	Left Cerebellum Cortex Volume, mm^3^		
	NC	MCI	SCI
Participants	9	9	10
Mean	52,724.6	50,075.8	51,631.8
Median	52,665.2	51,573.7	51,431.8
Std. Deviation	5596.8	4919.9	6323.4
Minimum	46,530.4	41,886.6	40,268.9
Maximum	62,715.6	57,555.6	62,242.0

**Table 4 brainsci-11-01134-t004:** Descriptive statistics for right cerebellum white matter volume in participants with normal cognition (NC), mild cognitive impairment (MCI), and moderate/severe cognitive impairment (SCI).

	Right Cerebellum White Matter Volume, mm^3^		
	NC	MCI	SCI
Participants	9	9	10
Mean	12,535.4	11,542.3	11,073.9
Median	12,279.9	11,124.0	11,237.1
Std. Deviation	1551.1	1542.9	2078.7
Minimum	10,220.2	9552.4	6591.0
Maximum	15,019.2	14,494.6	14,550.6

**Table 5 brainsci-11-01134-t005:** Descriptive statistics for right cerebellum cortex volume in participants with normal cognition (NC), mild cognitive impairment (MCI), and moderate/severe cognitive impairment (SCI).

	Right Cerebellum Cortex Volume, mm^3^		
	NC	MCI	SCI
Participants	9	9	10
Mean	53,667.8	51,066.6	52,981.4
Median	52,242.6	52,704.6	54,000.4
Std. Deviation	5982.1	5194.9	7410.6
Minimum	47,858.8	42,292.4	39,232.7
Maximum	63,872.6	58,125.1	65,617.1

**Table 6 brainsci-11-01134-t006:** Descriptive statistics for total cerebellar volume in participants with normal cognition (NC), mild cognitive impairment (MCI), and moderate/severe cognitive impairment (SCI).

	Total Cerebellum Volume, mm^3^		
	NC	MCI	SCI
Participants	9	9	10
Mean	131,612.0	124,113.8	127,193.8
Median	131,689.9	125,600.9	130,447.9
Std. Deviation	14,099.5	11,921.3	16,451.6
Minimum	115,429.9	104,217.5	93,977.8
Maximum	155,494.1	140,885.4	152,585.8

## Data Availability

The data presented in this study are available on request from the corresponding author. The data are not publicly available due to data privacy.

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
