# Peer review of "Cerebellar Cortex and Cerebellar White Matter Volume in Normal Cognition, Mild Cognitive Impairment, and Dementia"

_brainsci, 2021, doi:10.3390/brainsci11091134_

Round 1

Reviewer 1 Report

This manuscript presents 3T Magnetic Resonance Imaging (MRI) data in small sample of healthy controls, those with mild cognitive impairment (MCI), and those with moderate to severe (SCI) as determined by the MOCA (Montreal Cognitive Assessment).

The entire sample is very small (total n=28, with n=9 or 10 for each group).

There is no justification for using Freesurfer (which available evidence suggests does a poor job on cerebellar parcellation) than more cerebellar-focused parcellation methods

(e.g., volBrain Ceres: https://www.volbrain.upv.es

SUIT: http://www.diedrichsenlab.org/imaging/suit.htm).

Results are statistically insignificant.

English language requires review.

Control group is significantly younger than the MCI and SCI groups which is inappropriate given that MCI/SCI are age-related.

Introduction

There is evidence that the cerebellum connects with more frontal regions that the somatosensory cortex, references that should have been included to drive home the fact that the cerebellum is involved in cognition.

Methods

MOCA cut off scores are justified by only a single reference [17], do not comport with the general literature, and reduce the sample size (i.e., individuals with MOCA scores of 24-26 excluded).

Table 1 can be standardized by replacing 6 rows with only a single row for mean±SD (min/max) and median can be removed. Also, Table 1 does not include enough demographic data (e.g., sex, SES, education, other scores on neurocognitive tests, etc).

“Based on the neurological assessment and MRI patients in MCI and SCI had mixed type of cognitive impairment.” This statement is not clear. What types of cognitive impairments? Measured how?

Zero information on scan acquisition parameters. T1 or T2 weighted images?

Results

No justification for measuring right and left hemispheres separately.

Tables 2-6 can be standardized by replacing 6 rows with only a single row for mean±SD (min/max) and median can be removed and combining into a single table.

Discussion

Unusual presentation of discussion as bullet points.

Author Response

Dear reviewer,

Thank you for your time and comments. Here are answers to your comments.

Software for brain parcellation

As with every software available, there are pros and cons.

Our pros for FreeSurfer are that it is incorporated in our workflow (also for analyzing other brain structures) and gets regular updates (last update with stable v7.2 from 19th of July, 2021).

Regarding volBrain we have several concerns that the latest news is from year 2018, so we are not sure whether it is maintained and updated accordingly (comparing with Freesurfer there have been 4 releases with updates since 2020). Another concern is regarding data security. MRI data uploading to the web could be a serious data security threat and we can’t guarantee data security, thus we avoid this type of application.

We agree that more detailed cerebellar parcellation may show us additional results, but in this article, we focused on the gross anatomy of the cerebellum and analyzed cerebellar cortex volume and cerebellar white matter volume.

We will look at SUIT software and discuss how can we integrate it into our workflow and may use it for more detailed cerebellar parcellation in the future.

Participant age

As the control group age is younger, we would expect that there should be statistically significant differences between groups (taking into account cerebellar atrophy in aging, etc.), but still with this age difference we don’t see statistically significant differences. We also looked at our data excluding younger patients (to look at data with similar average age) and still did not find statistically significant differences.

Introduction

We added an additional paragraph that states cerebellar connection with prefrontal cortex and contribution to other pathologies (Autism and Schizophrenia).

MoCA cutoff scores

To avoid falsely classifying healthy individuals as cognitively impaired we excluded patients with MoCA tests scores 24, 25, and 26. While normative data of the MoCA test suggests the cutoff value of <26, according to Thomann et al. – “MoCA scores > 26 points may be considered as not pathological with very high accuracy, while scores ≤ 23 points are very likely, not healthy.” So, we can state with some certainty that participants included in the MCI group have pathological cognitive decline, and patients with MoCA scores 24, 25, and 26 will have an additional assessment in the future by neurologists.

Table design and additional data

We think that median values could be useful in a small cohort.

We added data in a table regarding patient sex.

Regarding education, we mentioned in the methods section “All participants had at least 16 years of higher education.”

We did not perform other cognitive tests to divide participants into specified groups, and we did not include participant socio-economic status in our analysis.

“Mixed type cognitive impairment”

In our participants, we did not have patients that had clear evidence of one type of dementia, i.e., vascular dementia, Alzheimer’s disease, Lewy body dementia, or frontotemporal dementia. All patients had at least some T2 white matter hyperintensities (according to Fazekas scale - 1st degree) and some degree of global cortical atrophy combination with other cerebral lobe atrophies.

Scan acquisition parameters

We included more detailed scan acquisition parameters (“3D T1-weighted MP RAGE sequence with 1 mm voxel size”). Our scan parameters are based on ADNI-3 MRI protocols.

Thank you for your time and have a nice day!

Best regards,

Nauris Zdanovskis

Reviewer 2 Report

This is a brief research report to deliver negative results concerning the correlation of cerebellar volume (on MRI) and cognitive performace as measured with MoCA. The main issues are raised in the research design. 

The total of 28 subjects are included, of them 19 suffer froom some level of cognitive impairment. First, it is critical to abandon the term "patients" when it comes to the description of the sample. Technically there are presented only 9 patients with confirmed severe cognitive impairment. Mild cognitive impairment is controversial category, which occupies the "transition" to disease borderline. Further authors are advised to expand their sample to at least 15 participants in each group, which is more likely to produce significant results.

Author Response

Dear reviewer,

Thank you for your time and comments.

We changed the term from “patients” to “participants” where it is appropriate. We understand and agree that the term “patients” in participants with normal cognition is inappropriate to use. Thank you for this comment!

While the mild cognitive impairment is controversial, according to Thomann et al. scores lower or equal to 23 are not healthy with 88% of accuracy, while the normative data of the MoCA test suggests the cutoff value of <26. To avoid controversial or false-positive cognitive impairment we excluded patients with MoCA tests scores 24, 25, and 26. So, we can state with some certainty that participants included in the MCI group have pathological cognitive decline, and patients with MoCA scores 24, 25, and 26 will have an additional assessment in the future by neurologists.

Although we plan to increase our participant count, now we are working with the data that we have. This was an exploratory study to assess whether cerebellar cortex and cerebellar white matter volume could be used as a biomarker. So far by analyzing literature and seeing our data we tend to think that it won’t be. Our research is consistent with described conclusions of Paradiso et al., Bernard et al., Hoogenda et al., and Mitoma et al.

Thank you for your time and have a nice day!

Best regards,

Nauris Zdanovskis

Round 2

Reviewer 1 Report

I still suggest as I previously had - that the authors combine tables 2-5. There is no justification for 4 separate tables and a single table will permit better comparisons for the readers.

Reviewer 2 Report

The authors have critically considered my comments in sufficient extent.